# Wind Tunnel Characterization of a Graphene-Enhanced PEDOT:PSS Sensing Element for Aircraft Ice Detection Systems

**DOI:** 10.3390/mi15020198

**Published:** 2024-01-28

**Authors:** Dario Farina, Marco Mazio, Hatim Machrafi, Patrick Queeckers, Carlo Saverio Iorio

**Affiliations:** 1Centre for Research and Engineering in Space Technologies (CREST), Department of Aero-Thermo-Mechanics, Université libre de Bruxelles, 1050 Brussels, Belgium; hatim.machrafi@ulb.be (H.M.); patrick.queeckers@ulb.be (P.Q.); carlo.iorio@ulb.be (C.S.I.); 2Department of Industrial Engineering, University Federico II of Naples, 80125 Napoli, Italy; marcomazio98@gmail.com; 3UFR Physique, Sorbonne Université, 75005 Paris, France

**Keywords:** ice formation, aircraft ice detection, artificial intelligence, hazard management, graphene sensors, in-flight safety technology, real-time ice monitoring, resistance signal analysis, PEDOT:PSS, wind tunnel

## Abstract

This study details the development and validation of a graphene-based ice detection system, designed to enhance flight safety by monitoring ice accumulation on aircraft surfaces. The system employs a semiconductive polymer (PEDOT:PSS) with graphene electrodes, interpreting resistance changes to detect water impact and ice formation in real time. The sensor’s performance was rigorously tested in a wind tunnel under various temperature and airflow conditions, focusing on resistance signal dependency on air temperature and phase change. The results demonstrate the sensor’s ability to distinguish water droplet impacts from ice formation, with a notable correlation between resistance signal amplitude and water droplet impacts leading to ice accretion. Further analysis shows a significant relationship between air temperature and the resistance signal amplitude, particularly at lower temperatures beneficial to ice formation. This underlines the sensor’s precision in varied atmospheric conditions. The system’s compact design and accurate detection highlight its potential for improving aircraft ice monitoring, offering a path toward a robust and reliable ice detection system.

## 1. Introduction

In-flight ice formation is a persistent issue for aviation safety that necessitates ongoing attention since it can reduce aircraft performance, compromise flight safety and increase operating expenses. A dangerous decrease in lift, an unwanted increase in drag, the beginning of flight-critical breakdowns as well as catastrophic events can easily result from the accumulation of ice on aerodynamically crucial surfaces such as wings, propellers and control components [1,2].

Over the years, the aviation industry has devised a variety of de-icing methods, both mechanical and chemical, to mitigate these risks. However, the inherently unpredictable nature of atmospheric icing requires the development of advanced detection systems that can prevent these hazardous conditions with greater accuracy and readiness [3,4].

Current methodologies for ice detection on aircraft are diverse, spanning from traditional mechanical systems to more sophisticated optical and acoustic sensors. For instance, mechanical detectors, the earliest form of ice detection, are based on the physical movement of a probe or surface to signal ice accumulation. These systems are straightforward but they often lack in sensitivity and promptness, both required for early detection which is crucial for enacting effective countermeasures [5,6].

The category of optical sensors utilizes infrared or laser technology to detect the presence of ice by measuring changes in light transmission or reflection. These sensors are known for their high sensitivity and can detect very thin layers of ice. Nevertheless, their effectiveness depends on unobstructed lines of sight and can be compromised by the presence of contaminants such as dirt or oil, which are common in operational aircraft environments [7].

Acoustic sensors represent another promising avenue, employing ultrasonic waves to detect changes in the material properties of the aircraft surface that occur with icing. These systems can be very precise but also complex, requiring calibration to account for the specific acoustic properties of the substrate and the different types of ice that can form [8,9].

Although these technologies represent significant advances in ice detection, each comes with its own set of challenges and limitations. Mechanical systems may fail to detect the onset of icing soon enough to activate de-icing systems promptly. Optical systems, despite being precise, are prone to false alarms and signal degradation under adverse weather conditions. Acoustic systems, in spite of their potential, are still in the early stages of practical application and often face challenges related to cost and integration into existing aircraft systems [10,11]. Recent advancements in anti-/de-icing aircraft sensors have seen significant developments, particularly in the application of advanced materials and sensing technologies. Notably, Caliskan and Hajiyev (2013) discuss various in-flight detection methods, including reconfigurable control systems and neural-network-based identification of icing conditions [12]. Additionally, the SENS4ICE EU project demonstrates progress in hybrid sensor architectures for enhanced ice detection [13]. These developments underline the evolving landscape of ice detection technology, setting the stage for our graphene-based sensor system’s potential contributions to flight safety in icy conditions.

The development of graphene-based sensors has paved the way toward new frontiers in ice detection technology. Graphene, a nanomaterial with outstanding electrical conductivity and mechanical strength, has the potential to revolutionize ice detection systems. Its high carrier mobility enables the detection of minor changes in resistance that could indicate the onset of icing [14,15].

This paper introduces a novel graphene-based sensor system, incorporating a PEDOT:PSS sensing layer, to detect ice formation on aircraft surfaces for prescribed ranges of environmental conditions.

The combination of the unique properties of graphene and the hygroscopic properties of PEDOT:PSS [16] can result in the detection of changes in electrical resistance that are indicative of ice forming.

The significance of this study lies in its potential impact on aviation safety by providing a sensor system that is more sensitive and responsive than current technologies. Such a system could greatly enhance the safety margins of aircraft operations, particularly in regions subjected to severe icing conditions [17,18].

By conducting thorough wind tunnel tests, this research aims at validating the performance of the graphene–PEDOT:PSS sensor system under controlled environmental conditions, focusing on variables such as temperature and airflow dynamics. The anticipated result is a robust set of data that delineates the responsiveness and reliability of the sensor, forming the design and implementation of future ice detection systems.

## 2. Experimental Setup and Methodology

This section outlines the structured approach undertaken to investigate the performance of a novel graphene-based ice detection system enhanced with a PEDOT:PSS sensing layer. The emphasis of this study is on the empirical relationships between the resistance signal of the detector and various environmental factors, such as air temperature, air speed and the phase state changes associated with ice crystallization.

The first subsections detail the synthesis and preparation of the PEDOT:PSS sensing films, elucidating the precise formulation and processing steps necessary to ensure their stability and sensitivity to environmental variables. The natural continuum is a description of the important contribution provided by graphene electrodes.

Subsequent sections will describe the specialized wind tunnel apparatus, which simulates the aerodynamic and thermal conditions encountered by the aircraft. Here, the sensor response to controlled variations in liquid water content (LWC), median volume diameter (MVD) of water droplets and water injection time are methodically assessed to deduce the critical parameters impacting water absorption by the sensing layer.

Furthermore, we explore the direct relationship between the amplitude of resistance change due to water exposure and subsequent ice formation creating maps that can guide users through the correct use of the sensor, and providing insights into the sensor’s ability to predict ice accumulation based on the initial water signal.

### 2.1. Fabrication of PEDOT:PSS Sensing Films

The fabrication of the PEDOT:PSS sensing films commenced with the preparation of the polymer solution. Clevios PH1000 from Heraeus Holding GmbH (Leverkusen, Germany), with a PEDOT to PSS ratio of 1:2.5, was filtered through a 0.45 µm polyvinylidene fluoride (PVDF) filter. To enhance the water stability of the polymer films, GOPS ((3-Glycidyloxypropyl)trimethoxysilane) was introduced into the solution to achieve a final concentration of at least 0.1%. The solutions were stirred for a minimum of one hour and finally sonicated for 5–10 min to ensure uniformity prior to film deposition [16]. PEDOT:PSS is based on a conjugated structure that enables charge mobility. Its charge transport mechanism is a blend of conduction along the polymer backbone and hopping between chains, where the corresponding electronic and ionic conduction are influenced by parameters like temperature and pressure [16].

### 2.2. Integration of Graphene as Electrode

Graphene is suitable for incorporation into complex three-dimensional structures thanks to its unique characteristics. Its many benefits include being lightweight, having low thermo-mechanical stress under heat cycling and its simplicity of integration into intricate geometries. Graphene, recognized for its superior electrical properties, significantly enhances the operational efficiency of our sensing system due to its excellent conductivity. This characteristic minimizes energy consumption, which is crucial for high-efficiency applications such as ice detection on aircraft. Its endurance and resistance to oxidation are further enhanced by its chemical inertness and resilience. Moreover, graphene is a perfect fit for a wide range of applications because of its smooth compatibility with a number of composite materials, including thermoplastics, glass-fiber-reinforced polymers and carbon-fiber-reinforced polymers.

Graphene’s very low and consistent resistance is especially useful since it allows its resistance values to be neglected from measurements when analyzing the resistance of PEDOT:PSS that predominates in the system. Graphene’s unique low and consistent resistance streamlines our analysis, enabling a more direct evaluation of resistive changes in PEDOT:PSS. In our experimental setup, we focused on a specific graphene electrode configuration, selected for its proven efficacy in ice detection on aircraft surfaces. Among several tested designs, the configuration illustrated in Figure 1a was chosen for its integration simplicity and effectiveness. This straightforward, straight-type electrode served as a standard against which other geometries were compared, providing a foundational benchmark for performance assessment. Figure 1b graphically depicts the integration of these electrodes into the ice detection system, illustrating both the chosen design and its practical application. Figure 1a details the physical layout of the straight-type electrode, while Figure 1b provides a system-level overview, including the sensing mechanism (PEDOT:PSS layer), the data acquisition hardware and the output interface, which displays the detected icing events. Beyond the electrodes’ shape, the graphene electrodes employed were glued in a polydimethylsiloxane (PDMS) substrate and appropriately shaped, and the distance between the electrodes was kept at 0.2 mm ± 0.05 mm. The graphene was fabricated in the form of ribbons of graphene GS50 by NANESA srl (Arezzo, Italy), with dimensions of 40 mm (length) × 5 mm (width) × 0.10 mm (thickness). In this study, GS50 refers to the graphene paper used as an integral component of our sensor design.

### 2.3. Sample Fabrication and Data Acquisition

The ice detection system’s efficacy rests on the precise fabrication of its sensor components. Central to the sensor’s design is the sensing element, composed of PEDOT:PSS enhanced with GOPS, as discussed in Section 2.1. This innovation underpins the sensor’s hygroscopic capability [16].

Carefully balanced constituents are decisive for signal fidelity. Pristine PEDOT-PSS is stored at 4 °C and filtered through a 0.45 µL PVDF filter during preparation. To achieve a homogeneous solution, the mixture is subjected to ultrasonication for 5 min. This preparation process ensures the material’s efficacy in water absorption, vital for operational success. The images of the sensor can be found in Appendix A, Figure A1.

Graphene paper is selected for its minimal sheet resistance and optimal balance of flexibility and durability, which are essential for reliable signal transmission. The intrinsic low resistance of graphene ensures that the detectable resistance changes emanate exclusively from the PEDOT-PSS response to the phase transition of water.

Silicone rubber, or PDMS, cut into specified dimensions, provides a flexible support base, essential for integration of the system into an airfoil. Graphene strips, prepared for precise measurements, form the conductive pathways. Copper tape secures the strips, facilitating robust electrical connections. A carefully deposited droplet of PEDOT:PSS, via drop-casting, bridges the strips, solidifying them into a conductive pathway upon drying. Afterward, the samples undergo a 24 h ambient drying phase, ensuring the stability of the conductive bridge. Resistance measurements validate the sensor’s stability before deployment.

The final step involves soldering wires to the highly conductive copper tape used as an interface between graphene and cables and interfacing the assembly with a data logger, the Agilent 34970A (Agilent Technologies, Santa Clara, CA, USA). This integration allows for real-time resistance monitoring and analysis, correlating resistance fluctuations with airflow-induced phase changes. The finalized sensors shown in Figure 2 are characterized by their flexibility, compactness and lightweight design. These attributes facilitate non-invasive installation on aircraft surfaces, enabling a network of sensors to function synergistically for localized ice detection. This approach is in line with the overarching objective of achieving energy efficiency through targeted de-icing, substantiated by precise detection mechanisms.

An important characteristic to highlight is the scalability of the sensor, with the possibility to resize based on the area of application but keeping the same capability of water and ice detection.

### 2.4. Facility Description and Operational Conditions

The tests were conducted in the low-speed wind tunnel at the Aero-Thermo-Mechanics (ATM) department of Université libre de Bruxelles in Belgium [19]. This closed-circuit facility allows for ice formation and accretion studies and is shown in Figure 3. The specifications of the wind tunnel are described in Table 1. In the wind tunnel, a camera, Jai 5000, recorded the entire process of the experiment to visually validate the ice formation at the same time that the signal variation makes its appearance. The main test section’s dimensions are 135 mm (height) × 180 mm (width) × 420 mm (length).

The operational conditions of the ATM wind tunnel are shown in Table 2.

The wind tunnel structure includes three main parts: instrumentation, cooling system and injection system. The instrumentation is composed of standard pressure and temperature sensors, a Pitot tube with Chromel–Alumel thermocouples for air velocity and a humidity sensor for measuring the relative humidity. Data visualization takes place through the Labview interface. The cooling system operates with air circulation through a heat exchanger, managed by a 10 kW refrigerating unit on the rooftop, with adjustments for thermal losses. The injection system is a two-loop system with a water circuit, a chiller, an external tank for the water–glycol mixture and an air circuit. Water pressure and droplets formation are regulated by the Model Delta Autojet 2008+ Spray Control panel (Spraying Systems Co., Wheaton, IL, USA). The nozzle selection plays a crucial role in droplet size control and is detailed in Table 3. Specifically, for the purpose of this paper, we used two nozzles, i.e., “SUJ1A” and “TPU 2505”. SUJ1A nozzles are particularly effective for simulating real-world supercooled conditions due to their low LWC and small droplet size. Although humidity was not directly controlled, the subfreezing temperatures and the nozzle’s distance from the sensor imitated realistic droplet impingement scenarios. Supercooling can be seen from the observed phase changes upon droplet impact, monitored by a top-positioned camera. The images can be found in Appendix A, Figure A2.

### 2.5. Analysis of Types of Signals

This segment discusses the signals acquired during wind tunnel tests that examine the behavior of a droplet’s impact detector. We describe the variation in the internal resistance of the detector, which helps to identify the collision of droplets with an airfoil and any subsequent phase change in the water absorbed. The wind tunnel tests are broadly categorized into three types based on the environmental conditions:Dry air conditions (resistance signals are presented in Figure 4);Water impact (resistance signals are shown in Figure 5a);Water impact and ice formation (resistance signals are shown in Figure 5b).

Dry tests are performed when both water injection and ice formation are absent. These tests aim to examine the detector’s behavior solely under varying air temperatures. A decreasing trend in resistance was observed with increasing temperature, which is consistent with the existing literature on PEDOT:PSS and graphene [16]. Figure 5 illustrates the relationship between the resistance and the temperature variation of the airflow in the wind tunnel. The study focuses on how water droplets impacting the sensing element lead to a significant drop in resistance, as illustrated in Figure 5a. These tests are crucial for assessing ice formation. Figure 5b demonstrates the detector’s resistance behavior under freezing conditions, highlighting its ability to accurately identify icing events on an aircraft’s airfoil. Even though the freezing process occurs in several seconds, the time lapse between the two signals is due to the charge transport mechanism of the PEDOT:PSS, mentioned in Section 2.1, needing a certain amount of time before leading to the characteristic peak of the resistance.

It should be noted that the background drift in resistance seen in Figure 5 is primarily due to the hygroscopic nature of the PEDOT:PSS layer, sensitive to ambient humidity and temperature changes. This sensitivity, as detailed in Dongo et al., 2023 [16], along with the material’s stabilization and aging, contribute to the resistance drift observed.

This relationship between resistance change and ice formation is key to the sensor’s effectiveness in real-time ice detection under dynamic flight conditions.

## 3. Results and Discussion

### 3.1. Amplitude Analysis and Correlation

In this study, we performed an in-depth analysis of the resistance signal variations due to ice formation on an airfoil surface during wind tunnel experiments. These dynamic tests built upon the static conditions described earlier in the text, but involved a sequence of water injection, ice accumulation under variable airflow conditions and a melting phase to reset for subsequent tests. The detector’s internal resistance exhibited an increase when ice formed, establishing a direct link between resistance changes and the water’s phase transition. Normalization of resistance signals was employed to mitigate variations inherent to individual detectors, utilizing the initial resistance value prior to the injection system’s activation for baseline establishment. This approach facilitated coherent comparisons across different test conditions. In the characterization tests, we saw that when a water droplet hits the sensor element, the resistance drops sharply, while during ice formation, the resistance increases. This trend remained consistent across varying air temperatures and injection parameters, with the amplitude of ice signals becoming more pronounced as the ambient air temperature decreased. Thus, the normalization for the water detection, $R_{N}^{\mathrm{water}}$, is defined as the minimum resistance value, $R_{\min}$, with respect to the initial one, $R_{i}$, while that for ice formation, $R_{N}^{\mathrm{ice}}$, is defined as maximum resistance, $R_{\max}$, with respect to the initial one:(1)$$\begin{array}{cc} R_{N}^{\mathrm{water}} & =\frac{R_{\min}}{R_{i}} \end{array}$$(2)$$\begin{array}{cc} R_{N}^{\mathrm{ice}} & =\frac{R_{\max}}{R_{i}} \end{array}$$

The amplitude of the ice signal, $\Delta R_{N}^{\mathrm{ice}}$, can be of interest and is given by
(3)$$\Delta R_{N}^{\mathrm{ice}}=\frac{R_{\max}-R_{i}}{R_{i}}=R_{N}^{\mathrm{ice}}-1$$

Table 4 and Table 5 present the parameters used in the text. Table 5 concisely summarizes the testing conditions for evaluating the ice detection sensor. It details experiments conducted across a range of airspeeds (20 to 60 m/s), air temperatures (−20 °C to 10 °C) and water droplet sizes (30 and 300 μm), with the number of experiments varying from 28 to 115 per condition set. The parameters for our experiments airspeeds from 20 to 60 m/s, air temperatures between −20 °C and 10 °C, and water droplet sizes of 30 and 300 μm—were chosen to mirror typical conditions during an airplane’s takeoff and initial climb. The selected droplet sizes are indicative of those encountered in ice-subposed environments like low-altitude clouds, ensuring our study’s relevance to real-world aviation scenarios. This table reflects the sensor’s comprehensive evaluation under diverse and realistic atmospheric conditions, ensuring a robust assessment of its performance and reliability. Table 5 outlines specific parameters for a subset of ice detection sensor experiments. It provides detailed insights into tests conducted at various airspeeds (20 to 60 m/s), corresponding Mach numbers (0.06 to 0.19), total air temperatures (TAT) ranging from −5 °C to −20 °C, a consistent median volume diameter (MVD) for water droplets at 30 μm and varying injection times (5 to 30 s). This matrix showcases the sensor’s examination under controlled conditions, highlighting its performance in response to changes in speed, temperature and duration of water exposure.

#### 3.1.1. Effect of Temperature

The major factor scrutinized is the effect of temperature on ice signals. Wind tunnel tests for this purpose were conducted with a constant airspeed, and the liquid water content and droplet dimensions were also maintained at constant levels. Experiments spanned a temperature range from −20 °C to 5 °C, with results portrayed on maps using spot markers. These maps represent the normalized ice signal amplitudes at specific temperatures for air velocities of 20 m/s, 30 m/s, 40 m/s and 60 m/s. Tests were conducted under fog (MVD = 30 µm, depicted by blue spots) and rain (MVD = 300 µm, shown by red spots) conditions. The dashed line on the maps segregates the regions where ice formation occurs (above the line) from the regions where it does not (below the line). According to Definition (Equation 3), such regions underneath the dashed line are characterized by a negative value of the ice signal’s amplitude since $R_{N}^{\mathrm{ice}}<1$ when ice formation does not take place. In fact, the maximum resistance (the maximum is taken after the resistance drop due to water impingement) in the case where no ice forms is lower than the initial one (resistance increase due to ice does not occur in that case).

Figure 6 depicts the wind tunnel tests focusing on the temperature’s effect at different airspeeds and under varied conditions. Table 6, Table 7 and Table 8 provide a detailed comparison of amplitudes across different temperature ranges for various airspeeds under fog conditions.

It can be noticed from the maps in Figure 6 that the ice signal amplitude amplifies as the air temperature decreases. This trend is consistent under both fog and rain conditions. Upon examining the amplitude values in Table 6, Table 7 and Table 8, under fog conditions across different airspeeds, a noticeable pattern emerges relative to the temperature ranges. At an airspeed of 20 m/s, the amplitude’s responsiveness to temperature is very strong. As temperatures reduce to the coldest range, between −20 °C and −15 °C, amplitudes surge to their highest values. As the cold relents slightly, moving toward warmer ranges, the amplitudes curtail systematically. Interestingly, once temperatures edge beyond the freezing mark of −5 °C, the amplitudes exhibit potential reversals, hinting at complex ice–temperature dynamics. When the airspeed increases up to 40 m/s, the response is quite different. Although the trend is quite similar, we can see that the dispersion in the data is smaller than at lower air speeds, with the difference being noticeable at the lower temperature ranges. The values of the amplitudes, between −15 °C and −5 °C, show less variance with the mean value.

A possible explanation might be that a higher air speed results in a higher heat transfer by convection, which facilitates the freezing process of the droplets that are absorbed by the sensor. The smaller dispersion is even more visible (especially at the temperature range −15 °C and −10 °C) at the higher airspeed of 60 m/s. For more details, see Appendix A, Figure A3.

To differentiate between temperature changes and water adsorption effects on our sensor’s resistance, we propose employing a differential measurement setup with dual sensors, one of which is moisture-shielded, to isolate temperature effects. Additionally, we suggest integrating advanced data processing algorithms, potentially machine-learning-based, for pattern recognition and response differentiation. Further, exploring material optimization and environmental compensation through supplementary temperature and humidity sensors could aid in selectively distinguishing these effects. This approach aims to enhance the sensor’s specificity, necessitating further research for effective implementation.

#### 3.1.2. Effect of Airspeed

In this section, we investigate the impact of wind speed on the amplitude of ice signals. The wind tunnel allows us to adjust the air velocity by altering the rotation speed of the compressor.

The correlation between air velocity and ice signals can also be verified from Table 9, Table 10 and Table 11 and the maps in Figure 7.

The air speed appears to influence the amplitude at lower temperature ranges of the air, while at higher temperatures, its impact is non-significant. In the configuration of the wind tunnel, it appears that the air flow with temperatures higher than −10 °C does not contain sufficient convectional energy to influence the freezing process.

In combination with the previous observations, when varying the air temperature at constant airspeed, we can see that there is an interplay between, respectively, the effects of the airspeed and air temperature on the behavior of the amplitude. The effect of the temperature is visible at higher airspeeds, whereas the effect of the airspeed is visible at lower temperatures.

The increase in airspeed has two effects. The first is to enhance the kinetic energy of impacting droplets. As a result, more droplets tend to glide over the surface without gathering at the leading edge. This leads to fewer absorbed frozen droplets. The second effect is that when the temperature is low enough, the droplets, upon touching the sensor, tend to absorb and freeze more easily. In that case, a higher convection (higher airspeed) actually favors more ice formation due to enhanced heat transfer. Thus, a combination of high airspeed and low temperature favors more ice formation, which is clearly detected by the sensor, as shown by the measured resistance amplitudes of the sensor.

For more details, see Appendix A, Figure A4.

#### 3.1.3. Effect of MVD

The quantity of liquid injected (LWC), the dimensions of the droplets (MVD) and the injection time are definitely responsible for ice detection effectiveness since the amount of water absorbed mainly depends on these parameters. The second parameter is discussed further in this paper, and two different cases are simulated depending on the orifice diameter of the nozzle chosen for the injection: very small droplets (MVD = ∼30 μm) and large droplets (MVD = ∼300 μm). The first case represents fog conditions, whereas the second case represents rain conditions. Figure 8 shows how the droplets’ dimensions influence the ice signal.

From the observations derived from the data represented in Figure 8, it can be observed that when the temperature is not low enough, as in Figure 8b, larger droplets are not able to freeze and run along the airfoil due to the airflow. This does not happen in fog conditions where droplets are small, and they manage to accumulate on the detector. Smaller droplets are also lighter, so they tend to follow the streamlines and assume the temperature of the air very quickly since the contact area is larger. Let us zoom further into this observation. Figure 9 and Figure 10 show signals of the resistance amplitudes at, respectively, two airspeeds, where each figure considers both fog and rain conditions. Signals reported in Figure 9a,b are obtained at 60 m/s at an air temperature of −4 °C. In fog conditions, ice formation can occur (recognized by the steep increase in the resistance amplitude), whereas there is no ice signal after injection in rain conditions. As mentioned above, this is due to the large diameter of the droplets injected. In Figure 10a,b, ice formation takes time since the temperature is higher, and water absorbed by the PEDOT:PSS does not immediately freeze. Nevertheless, the same observation can be made. While in fog conditions ice formation occurs, for droplets that have a larger volume, ice cannot form even after more than 5 min.

Finally, it appeared that the amplitude of the ice signal is linked to the amplitude of the water signal. The latter is proportional to the liquid water content being injected in the wind tunnel. Therefore, it has been observed that the bigger the decrease in the resistance at the injection (because of a larger amount of injected water), the bigger the increase in the resistance when ice formation occurs. For more details, see Appendix A, Figure A5.

## 4. Conclusions

This study establishes a correlation between the signals acquired by our detector and airflow conditions to enhance ice detection in aviation. Making use of materials science and sensor technology, we use a wind tunnel to evaluate the effectiveness of a graphene-enhanced PEDOT:PSS sensor. Our focus is on the sensor’s ability to accurately detect ice formation, considering variables like air temperature, droplet size and airspeed.

Essentially, it was observed that for a fixed airspeed and median droplet diameter, ice signal amplitudes increase by decreasing the temperature. In addition, lower values of the liquid water content result in a smaller amplitude of the ice signals for a prescribed airspeed and air temperature. The investigations highlighted that the acquisition of a specific ice signal’s amplitude in the function of a given type of simulation (e.g., in terms of injection parameters, airspeed and temperature) makes it possible to achieve a deterministic operation of the ice detection system. The most important points of such a cutting-edge technology are the compactness, lightness and flexibility of the detector used. These features definitely allow for a straightforward installation in several parts of an aircraft. The correlations presented in this paper between airflow parameters and the resistance’s trend reveal a correspondence between the signals acquired and the physics of the phenomena. This allows us to state that such a technology has great potential with possible future applications on the market in a more sustainable aviation prospective.

## 5. Patents

International patent n° WO2022162068A1/EP4036565A1, https://www.ulb.be/en/technology-sharing/device-and-method-for-detecting-ice-formation-technology-offer.

https://worldwide.espacenet.com/patent/search/family/074346829/publication/EP4036565A1?q=pn%3DEP4036565A1 (accessed on 4 December 2023).

## Figures and Tables

**Figure 1 micromachines-15-00198-f001:**
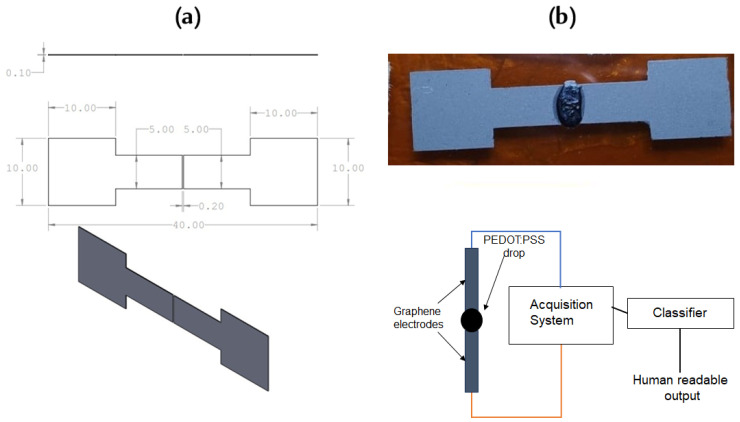
(**a**) Sketch of the electrodes designed and used in all characterization experiments. Dimensions are represented in millimeters. (**b**) The integrated system with graphene electrodes, PEDOT:PSS sensing layer, data acquisition system and readable output representation.

**Figure 2 micromachines-15-00198-f002:**
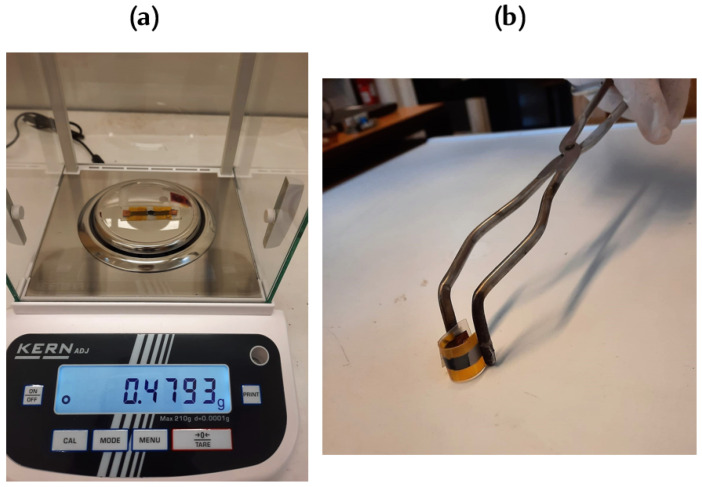
Features of the fabricated sample. (**a**) Sample weight demonstration. (**b**) Sample’s flexibility.

**Figure 3 micromachines-15-00198-f003:**
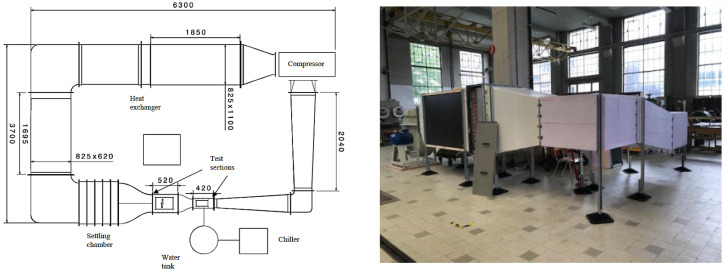
Wind tunnel planform and external view.

**Figure 4 micromachines-15-00198-f004:**
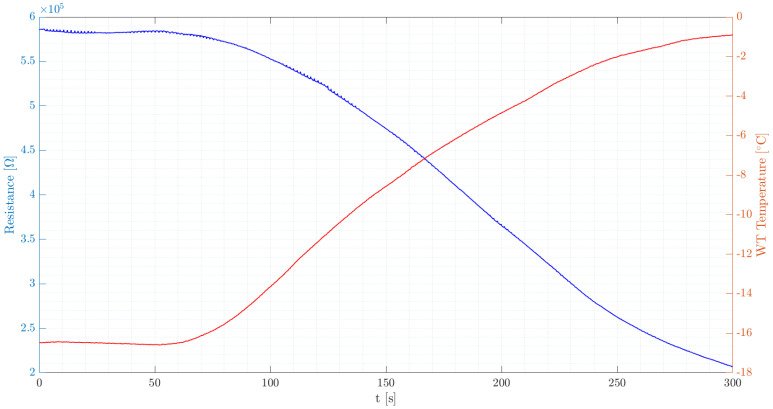
Behavior of the resistance value of the sensing element with the temperature variation of the dry airflow inside the wind tunnel.

**Figure 5 micromachines-15-00198-f005:**
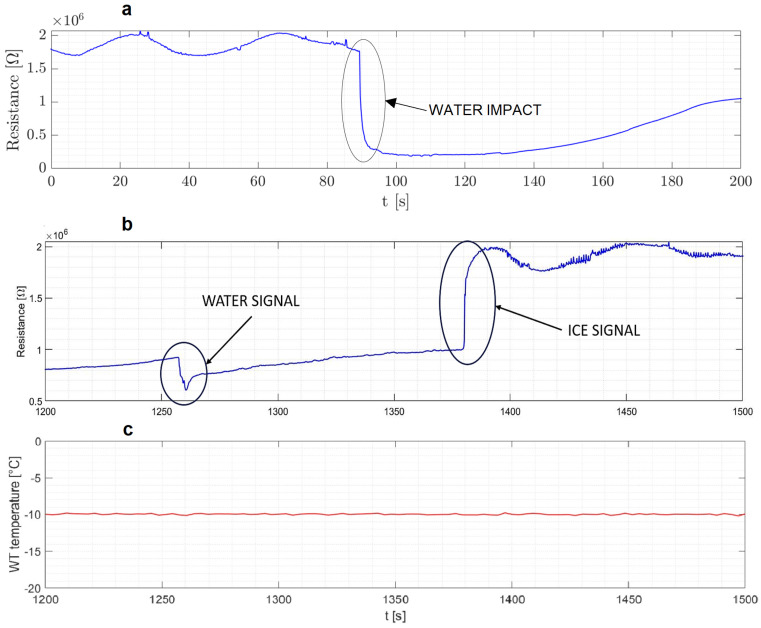
(**a**) Resistance pattern upon impact of water droplet on sensor before freezing occurs. (**b**) Resistance pattern involving both the droplet impact and the freezing of that droplet. (**c**) Temperature corresponding to the resistance pattern in (**b**), showing a steady temperature control during the freezing process. During the tests, the airspeed was 20 m/s, the liquid water content 1.5 g/m^3^ and the airflow temperature −10 °C.

**Figure 6 micromachines-15-00198-f006:**
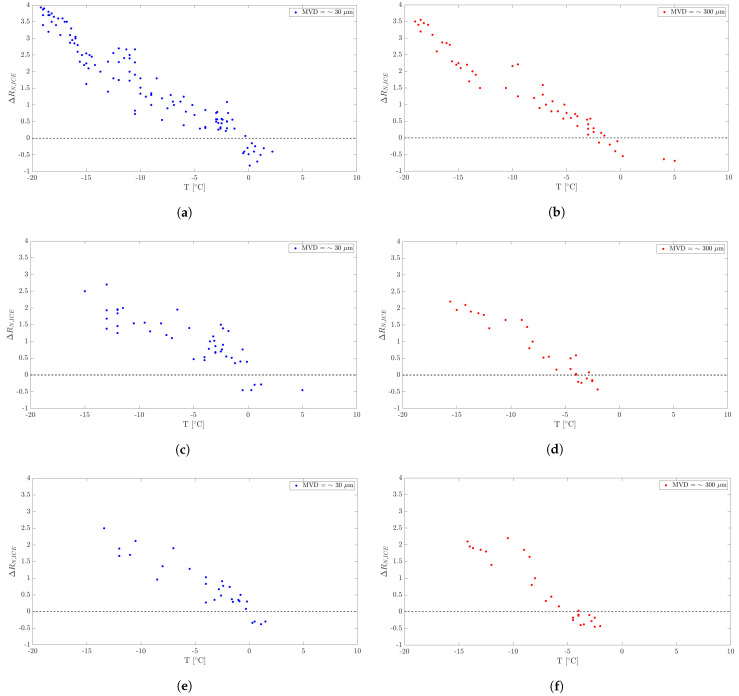
Wind tunnel tests. Effect of temperature on different airspeeds and conditions. (**a**) Airspeed = 20 m/s. Fog conditions. (**b**) Airspeed = 20 m/s. Rain conditions. (**c**) Airspeed = 40 m/s. Fog conditions. (**d**) Airspeed = 40 m/s. Rain conditions. (**e**) Airspeed = 60 m/s. Fog conditions. (**f**) Airspeed = 60 m/s. Rain conditions.

**Figure 7 micromachines-15-00198-f007:**
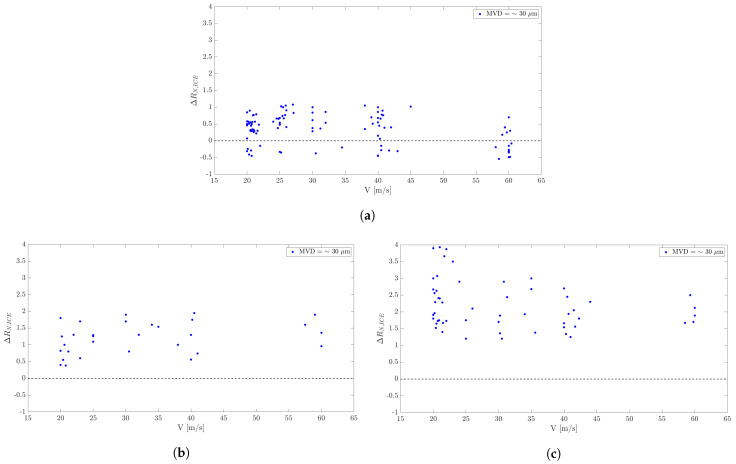
Wind tunnel tests. Effect of airspeed for prescribed temperature ranges in fog conditions. (**a**) T ≥ −5 °C. (**b**) −10 °C < T < −5 °C. (**c**) T ≤ −10 °C.

**Figure 8 micromachines-15-00198-f008:**
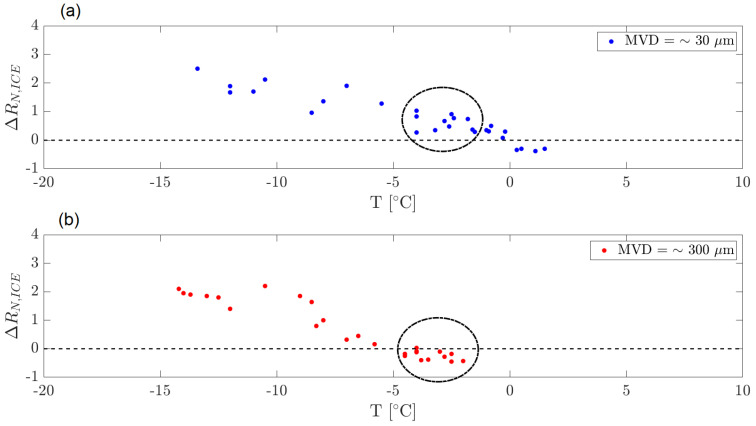
Wind tunnel tests. Airspeed = 60 m/s. Effect of droplets’ dimensions: (**a**) Fog conditions, (**b**) Rain conditions. The dashed circles illustrate, as an example, the favoured ice formation for smaller droplets with respect to the larger ones at the same temperature range.

**Figure 9 micromachines-15-00198-f009:**
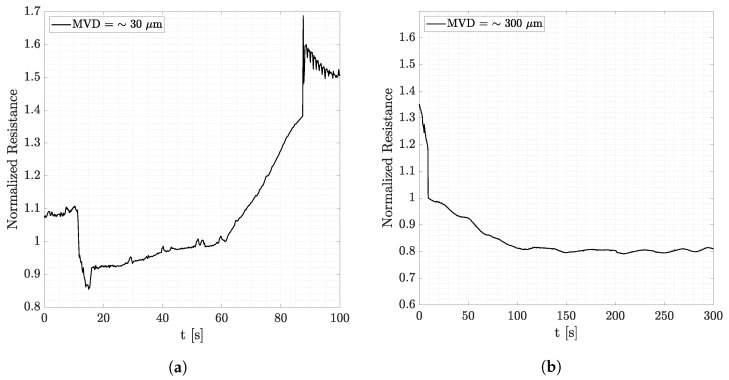
Wind tunnel tests at 60 m/s and T = −4 °C showing the effect of droplets’ dimensions under different conditions. (**a**) Airspeed = 60 m/s. T = −4 °C. Fog conditions. (**b**) Airspeed = 60 m/s. T = −4 °C. Rain conditions.

**Figure 10 micromachines-15-00198-f010:**
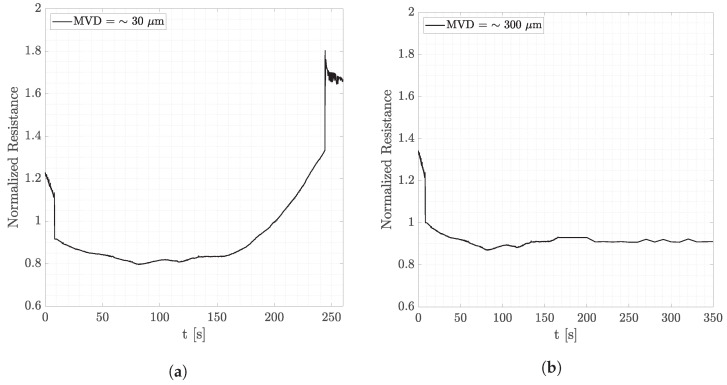
Wind tunnel tests at 25 m/s and T = −2 °C showing the effect of droplets’ dimensions under different conditions. (**a**) Airspeed = 25 m/s. T = −2 °C. Fog conditions. (**b**) Airspeed = 25 m/s. T = −2 °C. Rain conditions.

**Table 1 micromachines-15-00198-t001:** Experimental setup features.

Feature	Description
Construction	Closed-circuit, low-speed, galvanized steel ducts with large flanges and guide vanes.
Insulation	Silica aerogel for larger ducts, extruded polystyrene foam and cellulose wadding for complex shapes.
Airflow generation	11 kW centrifugal compressor delivering up to 9000 m^3^/h, powered by a DC motor and modulated via a frequency inverter.
Temperature range	Temperatures as low as −30 °C, using a heat exchanger system with R-410a coolant for efficiency and reduced CO_2_ emissions.
Test sections	Two test sections made of polycarbonate for clear visibility and excellent thermal insulation, accommodating various experimental setups.
Instrumentation	Standard pressure and temperature sensors, Pitot tube with Chromel–Alumel thermocouples, humidity sensor, and Labview system for real-time data.
Additional equipment	Flintec PC6 single-point load sensor for lift measurements and a high-resolution camera for ice formation observation.

**Table 2 micromachines-15-00198-t002:** ATM icing wind tunnel characteristics.

Feature	Specification
Maximum airspeed	90 m/s
Static temperature	Down to −20 °C
Liquid water content (LWC)	1.5 g/m^3^
Minimum median volume diameter (MVD)	30 µm

**Table 3 micromachines-15-00198-t003:** Nozzle specifications.

Denomination	Orifice Diameter (in)	Flow Rate (L/min)	MVD (µm)
SUJ1A	0.019	0.025	30
TPU 2505	0.056	2.0	300

**Table 4 micromachines-15-00198-t004:** Set of experiments performed.

Airspeed (m/s)	Air Temperature	MVD (µm)	Number of Experiments
20	From −20 °C to 10 °C	30 and 300	115
25	From −20 °C to 10 °C	30 and 300	55
30	From −18 °C to 10 °C	30 and 300	47
35	From −17 °C to 10 °C	30 and 300	28
40	From −15 °C to 10 °C	30 and 300	70
60	From −15 °C to 10 °C	30 and 300	52

**Table 5 micromachines-15-00198-t005:** Test matrix.

Airspeed (m/s)	Mach Number	TAT (°C)	MVD (µm)	Injection Time (s)
20	0.06	−5	30	5
20	0.06	−20	30	30
30	0.09	−15	30	30
60	0.19	−15	30	30

**Table 6 micromachines-15-00198-t006:** Amplitudes’ comparison. Airspeed = 20 m/s. Fog conditions. Effect of temperature.

Temperature Range	$\Delta R_{N}^{\mathrm{ice}}$
−20 °C < T < −15 °C	From 2.1 to 4.0
−15 °C ≤ T < −10 °C	From 0.8 to 2.7
−10 °C ≤ T < −5 °C	From 0.5 to 1.9
T≥ −5 °C	From −0.9 to 1.1

**Table 7 micromachines-15-00198-t007:** Amplitudes’ comparison. Airspeed = 40 m/s. Fog conditions. Effect of temperature.

Temperature Range	$\Delta R_{N}^{\mathrm{ice}}$
−15 °C < T < −10 °C	From 1.3 to 2.8
−10 °C ≤ T < −5 °C	From 1.1 to 2.2
T≥ −5 °C	From −0.4 to 1.5

**Table 8 micromachines-15-00198-t008:** Amplitudes’ comparison. Airspeed = 60 m/s. Fog conditions. Effect of temperature.

Temperature Range	$\Delta R_{N}^{\mathrm{ice}}$
−15 °C < T < −10 °C	From 1.6 to 2.5
−10 °C ≤ T < −5 °C	From 0.9 to 2.0
T ≥ −5 °C	From −0.4 to 1.1

**Table 9 micromachines-15-00198-t009:** Amplitudes’ comparison. T ≥ −5 °C. Fog conditions. Effect of airspeed.

Velocity Range	$\Delta R_{N}^{\mathrm{ice}}$
20 m/s < V < 30 m/s	From −0.5 to 1.2
30 m/s ≤ V < 40 m/s	From −0.4 to 1.3
40 m/s ≤ V < 60 m/s	From −0.5 to 1.3

**Table 10 micromachines-15-00198-t010:** Amplitudes’ comparison. −10 °C < T < −5 °C. Fog conditions. Effect of airspeed.

Velocity Range	$\Delta R_{N}^{\mathrm{ice}}$
20 m/s < V < 30 m/s	From 0.3 to 1.8
30 m/s ≤ V < 40 m/s	From 0.5 to 2.0
40 m/s ≤ V < 60 m/s	From 1.2 to 2.7

**Table 11 micromachines-15-00198-t011:** Amplitudes’ comparison. T ≤ −10 °C. Fog conditions. Effect of airspeed.

Velocity Range	$\Delta R_{N}^{\mathrm{ice}}$
20 m/s < V < 30 m/s	From 1.1 to 4.0
30 m/s ≤ V < 40 m/s	From 1.2 to 3.1
40 m/s ≤ V < 60 m/s	From 1.2 to 2.7

## Data Availability

All system routines are in the public domain and are available at the Université libre de Bruxelles.

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
