# Peer review of "Wind Tunnel Characterization of a Graphene-Enhanced PEDOT:PSS Sensing Element for Aircraft Ice Detection Systems"

_micromachines, 2024, doi:10.3390/mi15020198_

Round 1

Reviewer 1 Report

Comments and Suggestions for Authors

The authors present a method to detect the presence of water and ice on the surface of a flexible sensor, with the outlook of using the sensor on aircraft wings. The method makes use of a device based on a PEDOT:PSS active layer and graphene paper contacts. The data and the presentation are reasonable, although some minor tweaks are necessary in order to make the paper more comprehensible to the reader. Aside from the corrections outlined below, English should be improved. Recommendations follow:

There is no need to capitalize Liquid Water Content or Median Volume Diameter.

The sentence “Because of graphene’s strong conductivity, less energy is required, which leads to higher operational efficiency” is unclear. Less energy is required for what?

The caption of Figure 1a, or the sketch of the geometry itself, should contain units of length, i.e. millimetres.

It should be noted in the text, in Section 2.2, that GS50 is graphene paper, just to be clear what type of graphene is used in the study.

What is the origin of the background drift of resistance during the measurements shown in Figure 5? Authors should explain.

It is not clear under which conditions data shown in Figure 7 was taken. The axes, legends and captions on Figure 7b) and c) are identical. The difference should be explained in the caption of Figure 7.

There is significant cross-correlation between temperature and water adsorption on the sensor, i.e. in both cases the resistance measured across the sensor changes. Authors should discuss some potential strategies to implementing selectivity between temperature changes and water presence.

Comments on the Quality of English Language

The paper should be proof-read in order to fix strange formulation throughout the paper, which obscure the main scientific messages.

Author Response

Dear Reviewer 1,

Thank you for your insightful comments and suggestions on our manuscript. We have meticulously reviewed and addressed each point, Please see the attachment. This response, we believe, has greatly improved the clarity and comprehensiveness of our paper.

In line with your recommendations, we have also thoroughly proofread the manuscript to enhance the quality of the English language.

Your guidance has been invaluable in refining our work.

Best regards,

Dario Farina, Marco Mazio, Hatim Machrafi, Patrick Queeckers, and Carlo S. Iorio

Reviewer 2 Report

Comments and Suggestions for Authors

The authors present a study using innovative polymer materials to conduct an ice detection system. The topic is certainly interesting. However, some concerns must be addressed before it can published.

1) Please provide more background information about recently developed and equipped anti-/de-icing sensors on the aircraft. 

2) The abstract can be shortened into one paragraph instead of 5

3) The aircraft icing is directly related to the impingement of supercooled water and supercooled large droplets. The authors must provide more information to show that the droplets impacting the sensor in experiments reached the supercooled level.

4) In Fig. 5 b), there is a 120 s gap between the water droplet impingement and freezing. To the reviewer's knowledge, impingement and freezing are transient processes that happen in several seconds. The authors need to comment on this abnormal behavior.

5) Please explain more about the negative value of the amplitude of the ice signal.

6) Subtitles in Fig. 7 might be wrong.

Author Response

Dear Reviewer 2,

Thank you for your insightful comments and constructive feedback on our manuscript. We have carefully reviewed each of your points and provided detailed responses in the revised document.

To simplify the review process, we have combined the revised manuscript and our point-by-point responses to your comments into a single PDF file. 

The combined document is attached for your review. We hope that our revisions address your concerns effectively.

We appreciate your guidance and look forward to your feedback.

Best regards,
